# Appraising the Welfare of Thoroughbred Racehorses in Training in Queensland, Australia: The Incidence and Type of Musculoskeletal Injuries Vary between Two-Year-Old and Older Thoroughbred Racehorses

**DOI:** 10.3390/ani10112046

**Published:** 2020-11-05

**Authors:** Kylie L. Crawford, Anna Finnane, Ristan M. Greer, Clive J. C. Phillips, Solomon M. Woldeyohannes, Nigel R. Perkins, Benjamin J. Ahern

**Affiliations:** 1School of Veterinary Science, The University of Queensland, 4343 Gatton, Australia; s.woldeyohannes@uq.edu.au (S.M.W.); n.perkins1@uq.edu.au (N.R.P.); b.ahern@uq.edu.au (B.J.A.); 2School of Public Health, The University of Queensland, 4006 Herston, Australia; a.finnane@uq.edu.au; 3Torus Research, 4035 Bridgeman Downs, Australia; rmg@torusresearch.com.au; 4School of Medicine, The University of Queensland, 4006 Herston, Australia; 5Curtin University Sustainability Policy (CUSP) Institute, Curtin University, 6845 Perth, Australia; Clive.Phillips@curtin.edu.au

**Keywords:** racehorse, thoroughbred, epidemiology, musculoskeletal, injury, wastage

## Abstract

**Simple Summary:**

Musculoskeletal injuries (MSI) remain a concerning cause of racehorse morbidity and mortality with serious ethical and welfare consequences. Previous research examining risk factors for injuries report inconsistent findings. Age is thought to affect the risk of injury, but, to date, there have been no prospective studies specifically comparing injuries between two-year-old versus older horses. We aimed to: (1) determine the incidence of injuries for two-year-old and older horses, and whether this was affected by training track, season, or rainfall, and (2) determine the types of injuries affecting two-year-old and older horses, and whether horses trialled or raced after injury. Data were collected through personal structured weekly interviews with participating trainers over a 13-month period. Data were analysed using Poisson regression. The incidence of MSI in the current study was low (0.6%). The incidence of MSI in two-year-old horses was higher than older horses. Types of MSI varied between two-year-old and older horses and affected whether horses subsequently trialled or raced from 11 to 23 months after injury. A larger proportion of two-year-old horses had dorsal metacarpal disease and traumatic lacerations. A smaller proportion of two-year-old horses had suspensory desmitis, superficial digital flexor tendonitis, proximal sesamoid bone fractures, and fetlock joint injuries than older horses. Training track and rainfall did not affect the incidence of injuries. The season affected the incidence of injuries in two-year-old horses but not in older horses.

**Abstract:**

Musculoskeletal injuries (MSI) remain a concerning cause of racehorse morbidity and mortality with important ethical and welfare consequences. Previous research examining risk factors for MSI report inconsistent findings. Age is thought to affect MSI risk, but, to date, there have been no prospective studies comparing MSI in two-year-old versus older horses. This study aimed to: (1) determine the incidence of MSI for two-year-old and older horses, and whether this was affected by training track, season, or rainfall, and (2) determine the types of MSI affecting two-year-old and older horses, and whether horses trialled or raced after injury. A prospective survey was conducted with data collected through personal structured weekly interviews with participating trainers over a 13-month period. Data were analysed using Poisson regression. The incidence of MSI in the current study was low (0.6%). The incidence of MSI in two-year-old horses was higher than older horses (*p* < 0.001). Types of MSI varied between two-year-old and older horses (*p* < 0.001) and affected whether horses subsequently trailed or raced from 11 to 23 months after injury (*p* < 0.001). A larger proportion of two-year-old horses had dorsal metacarpal disease and traumatic lacerations. A smaller proportion of two-year-old horses had suspensory ligament desmitis, superficial digital flexor tendonitis, proximal sesamoid bone fractures, and fetlock joint injuries than older horses. Training track and rainfall did not affect MSI. The season affected MSI in two-year-old horses (*p* < 0.001) but not older horses. The major limitation was that trainers in this study were metropolitan (city) and our findings may not be generalisable to racehorses in regional (country) areas. Another significant limitation was the assumption that MSI was the reason for failure to trial or race after injury. In conclusion, the incidence of MSI was low in the current study and the types and the risk factors for MSI are different for two-year-old and older horses.

## 1. Introduction

Despite over three decades of active research, musculoskeletal injuries (MSI) remain a global problem for the thoroughbred (TB) racing industry [1,2,3]. There are important ethical, welfare, and economic consequences resulting from MSI. A principal issue is the serious injury and/or death of horses [4,5,6,7] and riders [8,9]. Musculoskeletal injuries are the most common cause of death, comprising over 70% of TB racehorse fatalities [10,11,12,13,14,15,16,17,18]. Previous studies report between 7% and 49% of race-day MSI resulting in death of the horse [4,7,13,19,20,21,22,23]. Furthermore, riders are more likely to be seriously injured or killed when their horse suffers a MSI [8,9].

The impact of MSI may be under-reported due to case definition. Studies with a case definition of race-day MSI will underestimate the problem because a large proportion of MSI and fatalities occur during training rather than during racing [10,11,21,24,25,26] or are only apparent once the horse has cooled down [11]. Furthermore, most MSI are repetitive stress injuries and the tissue damage occurs during training rather than on race day [27,28,29,30,31,32].

It has been proposed that two-year-old horses may be more susceptible to MSI due to their immaturity, even though there is little scientific information to support this claim. One study investigated reported a significantly higher incidence of carpal problems in horses that commenced training while the distal radial epiphyses were still open. However, this was not significant for injuries overall [33]. Other studies report age did not affect the incidence of MSI [5,17,34]. The inconsistent association between age and MSI may be due to variation in case definitions, study populations, and geographical locations, or a combination of these factors.

Study design and execution can also result in under-reporting or inaccurate reporting of MSI. Retrospective studies inherently contain errors due to incomplete or misclassified data [35,36]. Prospective studies potentially allow better data quality, but may be subject to attrition bias [35,37,38]. Relying on trainers to complete standardized questionnaires can also result in incomplete or inaccurate data [39,40,41]. Therefore, there is a need for prospective studies to accurately capture both training and racing data to better understand and reduce the impact of MSI.

Furthermore, there are no prospective studies that provide a detailed description of all injury types with most studies adopting a more broad outcome classification [11,42,43], or restricting their analysis to particular types of injury [11,26,44,45,46,47]. Similar reporting applies to studies of MSI in Thoroughbred horses racing over jumps [19,22,48] and Quarter Horse racing [14,15,45,49].

We address these data gaps through a prospective survey conducted over a 13-month period. Our aims were to: (1) determine the incidence of MSI for two-year-old and older horses and whether this was affected by training track, season, or rainfall, and (2) determine the types of MSI affecting two-year-old and older horses and whether horses trialled or raced after MSI.

## 2. Materials and Methods

### 2.1. Recruitment of Participants

A summary of the structure of thoroughbred racing in Queensland, Australia is provided in Appendix A
Figure A1. Briefly, thoroughbred racing in Queensland is divided into eight regions and each region is dividing into Totalisator Agency Board (TAB) and non-TAB racing clubs. The TAB racing clubs are licensed to conduct race meetings by the TAB. These races are televised and have a national wagering system. The non-TAB race meetings are not televised, and only on-course gambling is permitted. Metropolitan (city) TAB race meetings attract the highest prizemoney, which is followed by provincial and then regional (country) race meetings. The results of our pilot study, whereby we contacted the track supervisors of the Totalisator Agency Board (TAB) clubs in South East Queensland to determine the number of trainers and horses for each club during November 2017 are summarised in Appendix A
Figure A2. Trainers based at the three metropolitan racetracks administered by the Brisbane Racing Club (BRC) were invited to participate in the survey. These tracks (Eagle Farm, Doomben, and Deagon) comprise the major racetracks in South-East Queensland. Trainers exercised their horses at either Eagle Farm, Doomben, Deagon, or both Doomben and Eagle Farm.

Recruitment of horses was performed by recruiting trainers and enrolling all the horses under their care. The first author invited all licenced trainers at Eagle Farm, Doomben, and Deagon with three or more horses in work at the time of recruitment to participate in the study. A minimum of three horses was selected to ensure efficient data collection and trainer capacity to supply sufficient horses throughout the study.

### 2.2. Study Design

A prospective survey with the first author completing in-person structured weekly interviews over a 13-month period (56 weeks) with participating trainers or their forepersons. Details of the interview are described in Appendix A
Figure A3. Structured personal interviews enable clarification of any inconsistencies observed, ensuring accurate and complete data collection. Infrequently, when trainers or forepersons were not available for their scheduled in-person interviews, the interview was conducted by telephone, or rescheduled and conducted in-person within seven days. Data was also cross-checked with attending veterinarians when clarification was required. This situation did not commonly arise, as most trainers and forepersons had an excellent understanding of their veterinarians’ diagnoses.

### 2.3. Data Collection

The outcomes of interest were:(1)the total number of horses in training (number at risk of MSI)(2)the number of two-year-old horses in training(3)the number of new cases and the type of MSI and whether horses trialled or raced after injury(4)the number of fatalities

### 2.4. Total Number of Horses in Training

A horse in training was defined as a thoroughbred registered with the Australian Stud Book or equivalent international organization that was participating in race training under the care of the licenced trainer participating in the study. These horses were assumed to be healthy and capable of completing their training and racing exercise without clear lameness or medical conditions. The total number of horses in training were collected for each trainer every week. The unit of time at risk was one week. Horses were considered to be at risk for the week if they were in training for a minimum of five days. De-identified count data was collected rather than individual horse details.

### 2.5. Number of Two-Year-Old Horses in Training

A horse was defined as a “two-year-old” until 1 August of its third year of life. August first is the date where thoroughbred horses officially increase a year of age regardless of their actual date of birth. This definition incorporates all racehorses two years of age and younger, as racehorses are often younger than two years when they commence race training.

### 2.6. Number, Type of Musculoskeletal Injuries, and the Outcomes

A MSI was defined as any clinically relevant injury to the musculoskeletal system, incorporating orthopaedic and soft tissue injuries, which prevented the horse from training for 7 days. A 7-day period was chosen in order to be consistent with previous studies [11,40]. Our definition included any new MSI that occurred while the horse was in training, whether the actual injury was apparent during a race, during training, or was an accident in the stable. We also included osteochondritis dissecans, cervical stenotic myelopathy, and other developmental orthopaedic conditions if the horse initially was in training, appeared sound, and later progressed to a clinical lameness or gait abnormality that prevented the horse from training.

Musculoskeletal injuries were diagnosed by a veterinarian to avoid measurement and ascertainment bias. Horses in the study were under the close care of racetrack veterinarians registered in Queensland. Trainer permission was granted to obtain clarification from the attending veterinarian if required. A horse was defined as not trialling or racing again after an MSI if it did not participate in a trial or race for a minimum of 12 months after injury, which was confirmed by the Racing Australia (RA) public database [50].

### 2.7. Number of Fatalities

The total number of fatalities for the week and the reason were reported. A fatality was defined as any illness or injury resulting in the death of a horse or necessitating euthanasia on humane grounds.

### 2.8. Rainfall

The total rainfall for every month of the study period was obtained from the Bureau of Meteorology public database [51].

### 2.9. Data Analysis

Data analysis was performed using Stata 15.1^®^ (Statacorp, College Station, TX, USA). Continuous data were assessed for normality using histograms. Normally distributed data were presented as mean and standard deviation. Non-normally distributed data were presented as a median, interquartile range unless otherwise indicated. The weekly number of older horses in training was calculated by subtracting the number of two-year-old horses from the total number of horses. The median numbers of total horses, two-year-old horses, and older horses in training over 56 weeks were then calculated. The incidence of MSI was reported as weekly percentages of new MSI. Weekly percentages of MSI were calculated as the number of new cases of MSI divided by the number of horses in training (and, therefore, at risk of MSI) each week, multiplied by 100. The median percentages of MSI per week at risk over the 56 weeks were then calculated.

The weekly percentages of MSI for two-year-old and older horses were compared using the Wilcoxon rank sum test. Pearson’s Chi-squared test of proportions was used to compare the types of MSI between two-year-old and older horses, whether horses trialled or raced after injury, and whether the injuries were fatal. If the value in any cell was less than 5, Fisher’s exact test was used.

Poisson regression was used to assess the effect of training track, season, and rainfall on the number of counts of MSI, with the exposure variable showing the total count of horses at risk each week used as the offset to determine our incidence rate ratio. Analysis was performed to provide the Incidence Rate Ratio and the 95% Confidence Intervals for the counts of MSI observed in horses of all age groups, then repeated for MSI in older horses, and, finally, for two-year-old horses. Clustered models were used to correct for misspecification due to the known and unknown differences between trainers. Variables were considered in the multivariable model if the univariable Poisson model was significant at *p* < 0.2. Multivariable analyses were performed using a backward stepwise procedure to obtain a parsimonious model. Goodness-of-fit was tested using post-estimation deviance statistics and Pearson statistics. Significance was set at *p* < 0.05 for all tests.

## 3. Results

Forty trainers (15 at Eagle Farm, 6 at Doomben, 12 at Deagon, and 7 at both Doomben and Eagle Farm) were eligible for recruitment. Twenty-seven trainers (11 at Eagle Farm, 5 at Doomben, 4 at Deagon, and 7 at both Doomben and Eagle Farm) consented to participate. Details are shown in Figure 1.

Data were collected weekly over a 13-month (56 week) period from November 2017 to December 2018 for 26/27 (96%) of trainers. One trainer who did not complete the study contributed six months of data before retiring from training.

### 3.1. Incidence of MSI

There were a total of 202 MSI occurring in 195 horses. Ninety-seven of these 195 horses (50%) were older than two years and ninety-eight of these 195 horses (50%) were two years old. Seven horses (4%) experienced a second MSI during the study period. Three out of seven (43%) of these horses with a second MSI were older than two years and four out of seven (57%) were two years of age at the time of the first injury. Five out of seven horses (71%) with a second MSI had a recurrence of the original injury (carpal fragment or degenerative joint disease (n = 3), humeral stress fracture (n = 1), and dorsal metacarpal disease (n = 1)). The number of horses in training per week, the number of MSI, and the percentage of horses affected by MSI, stratified by age, is summarized in Table 1. The weekly percentage of two-year-old horses affected by MSI (median 1.3%, Interquartile range (IQR) 0.5%, 2.1%) was significantly higher than in older horses (median 0.3%, IQR 0.2%, 0.7%, *p* < 0.001).

### 3.2. Types of MSI

The types of MSI varied between two-year-old and older horses. A larger proportion of two-year-old horses compared with older horses had dorsal metacarpal disease and traumatic lacerations. A smaller proportion of two-year-old horses had suspensory ligament desmitis, superficial digital flexor tendonitis, proximal sesamoid bone fractures, and fetlock joint injuries than older horses (Table 2).

### 3.3. Outcome of MSI

Of the 195 horses with injuries, 82 horses (42%) did not trial or race again after injury. The follow-up period was between 12 and 23 months after injury. Of the horses that did not trial or race again after injury, 61/82 (74%) were older horses and 21/82 (26%) were two-year-old horses. Two-year-old horses were more likely to trial or race after MSI than older horses (77/98, 79% vs. 36/97 37%, *p* < 0.001).

Of the 202 MSI, horses did not race or trial after 83/202 (41%) of these. The older horse group sustained 61/83 (73%) of the injuries after which horses did not race or trial and the two-year-old group sustained 22/83 (27%).

The type of MSI also significantly affected whether horses trialed or raced again after injury. Horses were more likely to trial or race after dorsal metacarpal disease and traumatic injuries and less likely to trial or race after suspensory desmitis, superficial digital flexor tendonitis, and proximal sesamoid bone fractures (Table 3).

There were 14 fatalities during this 13-month study period of which most were due to musculoskeletal injuries (Table 4).

Twelve of 202 (6%) MSI were fatal in 12 horses and 6/12 (50%) of these were in two-year-old horses. The types of fatal injuries differed between two-year-old and older horses (*p* = 0.01). Humeral fractures were the most common fatal MSI 3/6 (50%) in two-year-old horses and sesamoid fractures were the most common fatal MSI in older horses 5/6 (83%). Most fatal MSIs occurred during training 9/12 (75%) rather than during a race 3/12 (25%). One two-year-old horse died after throwing the rider and galloping head-first into a barrier. Another two-year-old horse that was initially sound had to be euthanised due to rapid onset of severe neurological signs and cervical stenotic myelopathy being diagnosed.

### 3.4. Effect of Training Track, Season, and Rainfall

For horses of all ages, there was 1.6 times the rate of MSI in the summer (December to February) relative to winter (June to August) (*p* = 0.03). The track that horses trained on and the monthly rainfall did not affect the number of MSI (Table 5).

For older horses, neither the track trained on, season, or monthly rainfall affected the number of MSI in univariable models (Table 5). Season and rainfall were eligible for inclusion in the multivariable model. However, neither were significant. Therefore, the results of the univariable analyses were retained.

For two-year-old horses, there was 3.0 times the rate of MSI in the summer relative to the winter (*p* < 0.001). The training track and monthly rainfall had no effect on the number of MSI (Table 5). Monthly rainfall was eligible for inclusion in the multivariable model with season but was not significant. The results of the univariable analyses were retained.

## 4. Discussion

This is the first study to investigate the incidence of MSI in Queensland, Australia. All trainers meeting eligibility criteria at the major training tracks in Brisbane were invited to participate and 68% (27 of 40) agreed to be involved. Of these 26 (96%) participated throughout the longitudinal study period, reflecting the very level of enthusiasm and commitment. Participating trainers represented a convenience sample and, therefore, may not be representative of all trainers. A priori power analyses and sample size estimates were not performed in the design of this study because the study was descriptive in nature and because the design involved all eligible trainers being invited to participate and participating trainers enrolled all trained horses in the study. The high participation rate and very low drop-out rate provide increased confidence that the findings are likely to be generally representative of the population of racehorses in training in Queensland.

A small proportion (0.56%) of horses in training every week were affected by MSI. Comparing the impact of MSI between studies is problematic because of variation in outcome measures between studies. Most cross-sectional studies report prevalence (a “snapshot” of existing and new cases/number at risk), while cohort studies report either incidence rates (number of new cases/individual time at risk) or cumulative incidence (number of new cases/number at risk) [36,37,52]. Nevertheless, our findings are within the range of reported proportions of MSI from previous studies in other locations (0.28%–5.8%) [2,11,22,53,54]. The actual impact is likely to be lower than previous studies, as our case definition captured all MSI incurred during training, racing, and traumatic injuries. The proportion of fatal MSI in this study (6%) was also lower than studies from other regions [4,7,13,19,20,21,22,23].

Possible explanations for the low incidence of MSI and fatalities in this study include volunteer bias, close veterinary involvement, and active engagement of training modalities including water-walkers and treadmills. Trainers in this study all had ready access to veterinarians and diagnostic equipment, which is an important factor in monitoring horses and preventing MSI. Water-walkers and treadmills provide a short-term decrease in training intensity yet avoid the longer convalescent period and reduction in bone density associated with traditional paddock rest. Further investigation into the effect of water-walkers, treadmills, and equivalent training modalities on MSI is warranted.

In the current study, there was a higher incidence of MSI in two-year-old horses when compared to older horses. We suspect that the difference between two-year-old and older horses reflects two possible mechanisms of injury: (1) low training intensity in poorly adapted tissue, and (2) high training intensity in well-adapted tissue [43]. It was considered unlikely that trainers would subject horses with poorly adapted tissues to high intensity exercise. High speed exercise and training intensity is required for the tissue adaptation necessary to prevent MSI [55]. However, beyond a critical point, ongoing high intensity training results in tissue failure [56,57,58,59]. Two-year-old horses may have had insufficient high-speed exercise and training intensity to enable bone adaptation. Previous studies have also reported that two-year-old horses have a higher incidence of MSI [33,60]. There is a need for further research to investigate the reasons for two-year-old MSI and whether training practices can reduce the incidence of MSI. There is evidence that early training exercise may reduce the risk of MSI in two-year-old horses [61,62,63,64,65] and this possibility warrants further investigation.

Despite the low incidence of MSI in this study, the consequences for affected horses were severe. In fact, 12/195 (6%) of horses with MSI were euthanised or died and MSI accounted for 12/14 (86%) of the total deaths. In this study, eleven horses were euthanised on humane grounds due to the severity of injury and poor prognosis deemed by the attending veterinarian or consulting specialist surgeon. One horse died as a result of a traumatic head injury. This finding is consistent with other studies reporting MSI are the most common cause of death, comprising over 70% of TB racehorse fatalities [10,11,12,13,14,15,16,17,18]. Fatal MSI were evenly distributed between two-year-old (6/12, 50%) and older horses (6/12, 50%) in this study. Other studies have also found that the occurrence of fatal MSI was not affected by age [5,17,34].

The types of MSI differed between two-year-old and older horses. Two-year-old horses were more likely to sustain dorsal metacarpal disease and traumatic injuries and were more likely to trial or race again after injury than older horses. Dorsal metacarpal disease reflects poor adaptation of bone to withstand the normal stresses of racing [56], is sometimes considered a normal part of the process of training two-year-old horses [66], and has an excellent prognosis for full recovery. The trainer may be alerted to poor bone adaptation and interrupts or reduces the training program before further degenerative changes and more serious injuries develop. In contrast, older horses were more likely to sustain suspensory ligament desmopathy, superficial digital flexor tendonitis, proximal sesamoid bone, and fetlock joint injuries. They were less likely to trial or race again. These injuries are repetitive stress injuries [67,68,69,70,71,72,73] and are likely due to high intensity exercise in well adapted tissues that has exceeded the critical level.

The types of fatal MSI also differed between two-year-old and older horses and reflect different mechanisms of injury. Humeral fractures were the most common fatal MSI in two-year-old horses and these result from catastrophic progression of stress fractures [74]. Proximal sesamoid bone fractures were the most common fatal MSI of older horses and these result from excessive loading in well-adapted bones [75].

The training track did not affect the number of MSI for two-year-old or older horses. Previous studies examining the effect of the racetrack on MSI risk have reported inconsistent results [4,13,17,26,76,77]. Increased risk of MSI with track surface likely reflects high peak reaction forces due to hard tracks and high variance of peak reaction forces due to rough surfaces [78,79,80,81,82,83]. It is likely that track maintenance is more important than the actual track surface in determining the risk of MSI [26,81,82]. The lack of variation between tracks in the current study may reflect the intensive maintenance program for all BRC tracks (personal communication J. Rogers).

Seasonal variation strongly affected the number of MSI in two-year-old horses (three times the rate of MSI in the summer than in the winter) but not in older horses. The seasonal effect in this study was not due to rainfall as the total rainfall per month did not affect the number of MSI. We have two main hypotheses why two-year old horses experienced a higher number of MSI while older horses did not. First, the poorly adapted bones of two-year-old horses may be less resilient to the firmer track conditions in the summer [84]. Second, two-year-old horses were experiencing a rapid increase in training intensity in the summer, whereas older horses were not. Yearlings were experiencing their first serious gallops in the summer and the horses turning three the following August would have targeted the lucrative two-year-old races held in January. Our second hypothesis was based on consultation with industry experts and unpublished data (K. Crawford), as no published data exists.

The higher incidence of MSI in the summer for two-year-old horses is consistent with previous studies with a higher incidence of MSI in the summer in the USA [76] and a lower incidence of MSI or failure to finish a race in the winter in New Zealand [85,86]. However, those studies did not differentiate between two-year-old and older horses. The lower risk of MSI in the winter in New Zealand was attributed to the higher rainfall, heavier moisture content, and, therefore, softer going of the tracks [85,86]. There was no effect of rainfall on MSI in the current study. Other studies have also observed a lower risk of MSI on softer racetracks [7,13,19,77]. Alternatively, the lack of opportunity to gallop the horses due to the increased rainfall may be the reason for the lower risk of MSI in the winter in New Zealand. There are also no major racing events in New Zealand in the winter period and the reported reduced risk of MSI over the winter may, in fact, be due to fewer horses in training during this time. The lack of seasonal effect of MSI in older horses in the current study may reflect the opportunity to regularly gallop horses all year round. Furthermore, the number of horses in training does not drop over the winter, as the biggest racing events in Queensland are held during the winter carnival.

The predominant strength of this research was high quality data resulting from the unique access to a large number of trainers through personal weekly interviews over a 13-month period. Twenty-seven of 40 (67.5%) eligible trainers responded including 26 for the whole duration of the study. We feel this provided a representative sample of the horses of interest for the time period of the study. Personal interviews ensured that the data collected were both complete and accurate. Relying on trainers to complete standardized questionnaire results has resulted in incomplete data and the subsequent elimination of up to 29% of trainers from previous studies [39,40,41]. The prospective study design avoided the inherent errors due to incomplete or misclassified data that retrospective studies contain [35]. In fact, one study reported approximately 10% of cases being eliminated due to discrepancies in horse identity, race date, race location, race number, and race type [87]. A further study lost 87/301 (29%) horses that were either not in the database, had not had a timed workout recorded, or were in the first 30 days of their career or preparation [25]. Our weekly contact with trainers also avoided the potential for recall and attrition bias [35,37,38]. We attempted to minimize selection bias by enrolling all horses under the care of trainers rather than allowing trainers to nominate which horses would participate [38,88].

A further strength of this study was that case definition reflected the full extent of MSI, incorporating racing, training, and traumatic (accidental) injuries in training stables. The impact of MSI is often under-reported due to case definition and study design. Studies with a case definition of race day MSI will capture only a small component of the problem. There is good evidence that racing MSI are repetitive stress injuries and the damage occurs during training rather than during the race day [27,28,29,30,31,32]. Previous research has demonstrated that a large proportion of MSI and fatalities occur during training rather than during racing [10,11,21,24,25,26] or are only apparent once the horse has cooled down [11]. Our findings concurred that most fatal injuries (75%) occurred during training rather than on race day and that race day data under-reports the true prevalence of MSI.

Our inclusion and exclusion criteria were also developed to capture all MSI that potentially affect racehorses in training. For example, some studies have excluded osteochondral fracture fragments less than 5 mm from their case definition [21,22,89]. However, since our outcome was a clinically relevant injury rather than abnormal findings on survey radiographs, we included horses with fracture fragments of all sizes if they were diagnosed as the reason that the horse was unable to train. Excluding these fragments could significantly under-report the occurrence of MSI, as small carpal and metacarpophalangeal fractures are an important cause of morbidity [34,39]. Similarly, we included developmental diseases including osteochondritis dissecans (OCD) and cervical stenotic myelopathy (CSM) as, even though these conditions can be present with no clinical significance, they can also prevent a horse from training. Training practices could potentially influence whether these conditions progress to the point of clinical significance and prevent a horse from training.

Traumatic injuries (accidental injuries rather than injuries occurring during exercise) have also been excluded from the case definitions of previous studies [21,22,54,89]. Excluding traumatic (accidental) events from the case definition could also under-report the true incidence of MSI since traumatic events constituted 7% of the MSI in New Zealand horses [11]. We included horses with traumatic injuries in our case definition as they are an important cause of MSI and training practices could also potentially influence whether these injuries occur.

The main limitation of this study is that this study represents a subset of the Australian racing industry and our results are not necessarily globally applicable. Furthermore, there is inherent volunteer bias due to the voluntary nature of participation and self-selection of trainers [38]. The metropolitan trainers participating in our study may be more cognisant of MSI and have an interest in the outcomes of the study and training of elite horses. Our findings may not reflect horses from non-participating trainers or regional areas. Volunteer bias is a source of bias in most prospective studies investigating MSI in racehorses as they use a convenience sample of trainers deemed likely to complete the project [11,21]. Previous studies have reported the risk of MSI varies between trainers [22,26,39,40,54,85]. We adjusted for the known and unknown effects of the trainer by clustering data in the analysis. The Hawthorne effect, whereby people alter their behaviour or performance when they know that they are being observed could also potentially affect the validity of our study [37,88,90]. However, trainers were considered unlikely to change their practices during this study due to fear of decreasing their stables’ performance.

We also analysed count data rather than data for individual horse-time-at risk. This facilitated rapid population level data collection for a large number of horses. However, this method provides a less accurate measure of occurrence than collecting individual horse information [35]. Unfortunately, collecting individual level data for the entire study population was not feasible within the resources of the study. Furthermore, pilot studies indicated that trainers were not willing to commit the time that collecting individual horse data required.

Furthermore, there are serious limitations for assuming that MSI was the reason that horses did not trial or race after injury. Affected horses may have sustained paddock accidents during recuperation or returned to training but were subsequently retired for other reasons. In the current study, there were horses that did not race or trial following very minor injuries with a good prognosis including dorsal metacarpal disease and muscle injuries. It is unlikely that these injuries were the reason for retirement. Further scrutinization of the public database [91] and the Australian Thoroughbred Stud Book revealed one horse was exported to Singapore, and other horses had stewards’ warnings for unruly behavior and poor barrier manners. Long-term follow-up of MSI cases through personal interview is necessary to accurately determine the reasons that affected horses did not race or trial after injury. Unfortunately, this was beyond the resources of the current study. Finally, our dataset was small (median of 544 horses in training per week), the data collection period was only 13 months, and the follow-up period was only 12–23 months.

## 5. Conclusions

In conclusion, we reported a low incidence of MSI compared to previous studies and the reasons behind this finding warrants further investigation. The types of MSI, outcome, and risk factors for MSI vary between two-year-old and older horses. Consequently, future investigations into MSI should address these different populations of horses separately. A particular focus on two-year-old training practices is required since they are at higher risk of MSI and it is possible that early training exercise may decrease the risk of MSI. Thoroughbred racing is a highly contentious, high profile, and lucrative industry and, as such, attracts significant media and public opinion. It is vital that the industry is focused on understanding and mitigating the risks of MSI.6. Patents.

## Figures and Tables

**Figure 1 animals-10-02046-f001:**
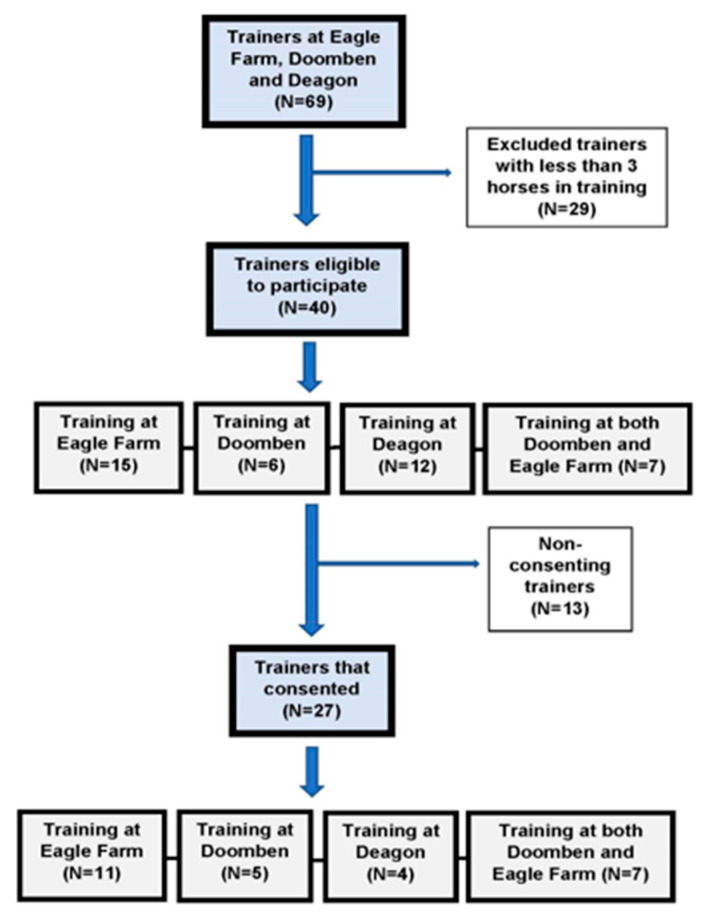
Recruitment of trainers to investigate the incidence of musculoskeletal injuries in thoroughbred racehorses in South-East Queensland. N = the numbers of trainers at recruitment time at the start of November 2017.

**Table 1 animals-10-02046-t001:** The number of thoroughbred racehorses in training, the number of musculoskeletal injuries (MSI) per week, and the percentage of racehorses in training sustaining musculoskeletal injuries per week, stratified by age and training track.

Number of Horses in Training per Week				Number of MSI per Week			% Horses with MSI per Week		
Horses of all Ages	Median (IQR)		Range	Median (IQR)		Range	Median (IQR)		Range
Eagle Farm	383	(381, 384)	375–388	2.5	(1, 4)	0–10	0.7	(0.3, 1.0)	0–2.7
Doomben	45	(43, 47)	38–52	0	(0, 1)	0–1	0	(0, 2.0)	0–2.6
Deagon	57	(54, 58)	49–60	0	(0, 1)	0–2	0	(0, 0)	0–3.7
Doomben and Eagle Farm	60	(57, 62)	50–69	0	(0, 1)	0–2	0	(0, 1.6)	0–3.8
Total	544	(538, 547)	530–559	3	(2, 5)	0–11	0.6	(0.4, 0.9)	0–2.0
Horses > 2 years old									
Eagle Farm	288	(265, 311)	219–329	0	(1, 2)	0–5	0.4	(0, 0.7)	0–1.7
Doomben	37	(34, 40)	30–43	0	(0, 0)	0–1	0	(0, 0)	0–3.0
Deagon	45	(42, 48)	39–56	0	(0, 0)	0–1	0	(0, 0)	0–2.3
Doomben and Eagle Farm	45	(38, 49)	30–54	0	(0, 0)	0–2	0	(0, 0)	0–5.0
Total	405	(380, 444)	331–473	1	(1, 3)	0–7	0.3	(0.2, 0.7)	0–1.6
2-year-old horses									
Eagle Farm	95	(71, 118)	56–165	1	(1, 2)	0–5	1.3	0.6, 2.4	0–5.7
Doomben	8	(6, 11)	2–14	0	(0, 0)	0–1	0	(0, 0)	0–50
Deagon	10	(8, 15)	0–19	0	(0, 0)	0–2	0	(0, 0)	0–16.7
Doomben and Eagle Farm	15	(12, 20)	9–27	0	(0, 0)	0–2	0	(0, 0)	0–10.5
Total	130	(98, 163)	71–210	2	(1, 3)	0–5	1.3	(0.5, 2.1)	0–4.3

**Table 2 animals-10-02046-t002:** The types of Musculoskeletal injuries that prevented thoroughbred racehorses from training for at least seven days and a comparison of the distribution between two-year-old and older horses. Variables significant at *p* < 0.05 indicated in bold.

Type of Musculoskeletal Injury Preventing Training for ≥ 7 Days	Number in Horses > 2 Years Old (% of Total)	Number in Two-Year-Old Horses (% of Total)	*p*-Value
**Dorsal metacarpal disease**	**5/47 (11)**	**42/47 (89)**	**<0.001**
Carpal fragment or degenerative joint disease	17/31 (55)	14/31 (45)	0.4
**Suspensory,** **Accessory of the deep digital flexor tendon** **, or** **Sesamoidean** **desmitis**	**18/22 (82)**	**4/22 (18)**	**0.001**
**Superficial digital flexor tendonitis**	**14/18 (78)**	**4/18 (22)**	**0.01**
**Traumatic laceration or injury**	**3/14 (21)**	**11/14 (79)**	**0.03**
**Proximal sesamoid bone fracture,** **Second/Fourth metacarpal/tarsal bone** **fracture or** **Sesamoiditis**	**10/12 (83)**	**2/12 (17)**	**0.02**
**Fetlock fragment, degenerative joint, or palmar osteochondral disease**	**9/10 (90)**	**1/10 (10)**	**0.01**
Ilial fracture, stress fracture, sacroiliac injury, or tibial stress fracture	6/10 (60)	4/10 (40)	0.4
Unknown	3/10 (30)	7/10 (70)	0.3
Muscular injury or Rhabdomyolysis	4/8 (50)	4/8 (50)	0.6
Humeral fracture or stress fracture	3/7 (43)	4/7 (57)	0.5
Quarter crack, Distal phalanx wing, or solar margin fracture, Pedal osteitis	5/6 (83)	1/6 (17)	0.1
Dorsal third metacarpal bone or condylar fracture	2/4 (50)	2/4 (50)	0.7
Osteochondritis dissecans, cervical stenotic myelopathy	1/3 (33)	2/3 (67)	0.5
Total	100/202 (50)	102/202 (50)	

**Table 3 animals-10-02046-t003:** The types of musculoskeletal injuries that prevented thoroughbred racehorses from training for at least seven days and a comparison of whether horses trialed or raced after injury. Variables significant at α = 0.05 indicated in bold.

Type of Musculoskeletal Injury Preventing Training for ≥ 7 days	Number of MSI that Horses did not Trial or Race after (%)	Number of MSI that Horses Trialled or Raced after (%)	*p*-Value
**Dorsal metacarpal disease**	**5/47 (11)**	**42/47 (89)**	**<0.001**
Carpal fragment or degenerative joint disease	16/31 (52)	15/31 (48)	0.2
**Suspensory,** **Accessory of the deep digital flexor tendin** **, or** **sesamoidean** **desmitis**	**15/22 (68)**	**7/22 (32)**	**0.01**
**Superficial digital flexor tendonitis**	**13/18 (72)**	**5/18 (28)**	**0.01**
**Traumatic laceration or injury**	**2/14 (14)**	**12/14 (86)**	**0.03**
**Proximal sesamoid bone fracture, splint fracture, or sesamoiditis**	**9/12 (75)**	**3/12 (25)**	**0.02**
Fetlock fragment, degenerative joint, or palmar osteochondral disease	4/10 (40)	6/10 (60)	0.6
Ilial fracture, stress fracture, sacroiliac injury, or tibial stress fracture	3/10 (30)	7/10 (70)	0.4
Unknown	4/10 (40)	6/10 (60)	0.6
Muscular injury or rhabdomyolysis	2/8 (25)	6/8 (75)	0.3
Humeral fracture or stress fracture	3/7 (43)	4/7 (57)	0.6
Quarter crack, distal phalanx wing, or solar margin fracture, pedal osteitis	4/6 (67)	2/6 (33)	0.2
Dorsal third metacarpal bone or condylar fracture	2/4 (50)	2/4 (50)	0.6
Osteochondritis dissecans, cervical stenotic myelopathy	1/3 (33)	2/3 (67)	0.6
Total	83/202 (41)	119/202 (59)	

**Table 4 animals-10-02046-t004:** The causes of death in 14 thoroughbred racehorses training and racing in South East Queensland.

Cause of Death	Number of Horses (%)
Proximal sesamoid bone fracture	5 (36)
Humeral fracture	3 (21)
Traumatic injury	1 (7)
Ilial fracture	1 (7)
Third metacarpal bone fracture	1 (7)
Cervical stenotic myelopathy	1 (7)
Total Musculoskeletal injuries	12 (86)
Colitis	1 (7)
Unknown, inconclusive post mortem findings	1 (7)
Total	14 (100)

**Table 5 animals-10-02046-t005:** The effect of training track, season, and rainfall on the number of musculoskeletal injuries in thoroughbred racehorses training and racing in South-East Queensland.

Horses of all Ages	Univariable IRR ^†^ (95% CI)	*p*-Value
Track		
Eagle Farm	ref	*
Doomben	0.84 (0.35, 2.01)	0.7
Deagon	0.68 (0.42, 1.09)	0.10
Doomben and Eagle Farm	0.85 (0.47, 1.53)	0.6
Season		
Winter	ref	*
Spring	0.98 (0.67, 1.44)	0.9
**Summer**	**1.61** **(1.06, 2.46)**	**0.03**
Autumn	0.9 (0.55, 1.50)	0.7
Rainfall		
Total rainfall (mm)	1.00 (1.00, 1.00)	0.8
Horses > 2 years old	Univariable IRR (95% CI)	*p*–Value
Track		
Eagle Farm	ref	*
Doomben	0.87 (0.79, 2.50)	0.3
Deagon	0.62 (0.29, 1.30)	0.2
Doomben and Eagle Farm	1.28 (0.63, 2.60)	0.5
Season		
Winter	Reference	*
Spring	0.75 (0.51, 1.12)	0.2
Summer	0.99 (0.43, 2.37)	>0.9
Autumn	0.62 (0.36, 1.05)	0.08
Rainfall		
Total rainfall (mm)	1.00 (0.99, 1.00)	0.2
Two-year-old horses	Univariable IRR (95% CI)	*p*-Value
Track		
Eagle Farm	Reference	*
Doomben	0.73 (0.32, 1.67)	0.46
Deagon	0.81 (0.53, 1.22)	0.31
Doomben and Eagle Farm	0.47 (0.17, 1.27)	0.11
Season		
Winter	ref	*
Spring	1.73 (0.83, 3.61)	0.15
**Summer**	**3.00 (1.89, 4.77)**	**<0.001**
Autumn	1.32 (0.65, 2.68)	0.37
Rainfall		
Total rainfall (mm)	1.00 (1.00, 1.00)	0.08

^†^ IRR = Incident rate ratio. CI = Confidence Interval Univariable Poisson regression models shown. Variables significant at *p* < 0.05 indicated in bold. Winter = June to August, Spring = September to November, Summer = December to February, Autumn = March to May. * not applicable as reference category.

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
