# Peer review of "Appraising the Welfare of Thoroughbred Racehorses in Training in Queensland, Australia: The Incidence and Type of Musculoskeletal Injuries Vary between Two-Year-Old and Older Thoroughbred Racehorses"

_animals, 2020, doi:10.3390/ani10112046_

Round 1

Reviewer 1 Report

Overall:  Excellent article. There are a few sections that require additional clarification in order for the reader to understand the process or outcomes reported.

Simple Summary

Make sure to include “MSI” after the term Musculoskeletal injuries is first mentioned in simple summary.  This is done in the abstract so it may not be needed in the simple summary unless it is to be shown to a larger audience.

Abstract

Good

Introduction

May want to include a discussion about how type or frequency of training may or may not be taken into account in various previous studies or if similar studies have been done on non-racing horses (or other breeds in the racing industry). 

Materials and Methods

Participants

              The places and locations are mentioned, but the actual number of horses considered in this study should be included in this section with adequate descriptions of the number of horses in each age group for each location.            

              Data Collection

              The number of trainers should be listed as well as if the interviews were conducted over the phone, in person, or reported through some other means.  Also mention if any other qualitative measures were taken into account for each interview (when was information on quality of veterinary care taken into account or the resources available?).  Were qualitative questions standardized?  If so, this needs to be listed in the article or appendix.  How often were these additional criteria collected? 

              Data Analysis

              I am happy to see that the authors/researchers chose to look at both normally and non-normally distributed data and took care to analyze these independently.    

Results

              In addition to the overview of younger horses and the results related to MSI for this group, the first paragraph should also give an overview of what was found for the older group of horses as well, including the same breakdown of injuries and percentages.  Although this information is broken down in the table, an overview is needed for the older horses for comparison within the text. 

Discussion

Lines 271-276 – Clarify the cause of death and access to rehabilitation methods.  At present, the paragraph mentions that MSI was the cause of death but previous sections mentioned the choices and care of veterinarians and later sections mention ability to recover.  Please specify why certain MSIs result in death (choice of individual veterinarian, lack of resources, etc).  Although this information may not have been available to the researchers, it should be mentioned. 

Lines 277 – 287 – Good description of the types of injuries, how they occur, and what they mean for each age group in terms of short and long-term consequences. 

The authors incorporate good thoughts as to the differences in MSI between younger and older horses regarding seasonal variations. 

The authors mention the Hawthorne effect, but could also mention reporting bias due to (potential) risk of cultural or industry pressure to conform to certain standards.  This should also be clarified if the authors feel this is a potential reporting factor (it is a problem in other countries, but may not be the case for New Zealand). 

Reviewer 2 Report

This paper describes musculoskeletal injuries in a small convenience sample of Australian Thoroughbred racehorses. I think it would be appropriate to include ‘Australian’ in the title of the paper because the results are not necessarily transferable to other populations of Thoroughbred racehorses worldwide.

For a non-Australian, who is familiar with management and training regimes of Thoroughbred racehorses in Europe and the USA, but not in Australia, I felt that it would be helpful for the reader to have a bit more background on management and training regimes of Thoroughbred racehorses in order to be able to put the results into context.

I realise that the data were collected from a convenience sample of trainers, however the overall number of horses is quite small and they were only followed for 13 months. Moreover, there was a big difference in the numbers of horses ≤ 2 years of age versus older horses.  Were power calculations performed to determine the appropriate numbers of horses from which to draw statistically valid conclusions?

The authors conclude that they reported a low incidence of MSI. Low with respect to what?

What are the authors conclusions about the ethics of racing 2 year olds?

Abstracts

Line 48 What does metropolitan mean in this context?

Line 49 What does 'regional areas' mean in this context?

The study is limited to a very small cohort of horses in Australia

Line 51 there is a rogue '.' after the

Line 53 The Keywords are supposed to provide additional words to those in the title, for use by search engines

Introduction

Line 181 What were your hypotheses & why?

Materials and Methods

Line 84 How was the number of participating horses determined and balanced between horses of 2 years of age versus older horses?

Power calculation?

Line 93 Please provide copy of  the structured interview

Line 111 Do you mean anonymised data were collected? How is ‘count data’ defined?

Line 126 So does this imply that all trainers consult a veterinarian and a horse never has 7 days off due to lameness without a veterinarian being consulted?

Line 130 If the definition of not trialling or racing was a period of 12 months off,  why  was the follow-up period only 11 months for some horses (line 199)?

Line 144 The median numbers .... were  

or

The median number   ....was

Line 146 So if a horse had for example a mild suspensory ligament injury, had time off, resumed work and then sustained recurrent injury is this classified as a new MSI? Please clarify.

Line 155 What does 'with an offset of the total count' mean?

Lines 159 & 162 Why α and not p, and why α=0.05 rather than < 0.05? Confusing given that P values are given in the tables.

Results

In order to have some idea how to interpret this data we need to know how Australian racehorses are managed and trained; track surfaces, track maintenance etc.

line 190 compared with older horses

Line 191 suspensory ligament  desmitis is tautology

Line 185 why do you refer to α here but give P values in the table?

In Tables 2  & 3 correct Nomina Anatomica Veterinaria nomenclature should be used

e.g., accessory ligament of the deep digital flexor tendon rather than 'check'

Sesamoidean is misspelt

Rhabdomyolysis is misspelt

Third phalanx should be distal phalanx

Line 207 tautology

Line 227 How are summer and winter defined? Your audience is global!

Table 5 CI is not defined nor mentioned in the Materials and Methods

Discussion

Lines 261-3 It is not clear what the authors are trying to say here. Why couldn't high training intensity in poorly adapted tissue be important?

Line 270 This would not be consistent with the results of  the studies of Elwyn Firth's group ( the global consortium – McIlwraith, Smith, van Weeren et al.)

Line 271 How does this incidence of MSI in horses of ≤2 years of age in the current  study compare with other studies?

How does the spectrum of injuries seen in the young and older horses compare with other studies?

Line 286 another rogue '.'

Line 290 The statement about PSB fractures would be completely contrary to the results of Sue Stover's group  (California)

Line 305-7 Was work intensity monitored? If not why not?

Lines 323-5 The readership need to know about the structure of the interviews

Line 325 data...were

Line 362 Other limitations:

Small data set

Limited long-term follow-up

Only 1 year of data

Subset of the Australian racing industry – results not necessarily globally applicable

Line 383 Please write DMD in full, as elsewhere in the manuscript

Please check the references carefully. The citation of journals etc is inconsistent.

Equine Vet J,

Equine Veterinary Journal

Lines 445, 464, 496, 580, 595 The references are incomplete

Line 500 Incorrect citation

Line 511 What is this?

Line 533  available from????

Lines 547, 584 Which Supplement number and which volume?

Round 2

Reviewer 2 Report

Referee’s report

The manuscript is improved, but some errors persist that require change.

Line 85 You need to state your hypotheses and do sample size calculations. You have said in responses to reviewers

‘We hypothesised that two-year old horses would have a higher incidence of MSI, but these would be less severe. This hypothesis was based on extensive industry experience at a number of levels.’

OR

The authors have stated in their response to reviewers 

 ‘This study was an initial survey to estimate the incidence and type of MSI in South-East Queensland horses, and not a hypothesis driven study. Thus, we did not estimate sample size a priori, but attempted to enrol the maximum number of respondents. Twenty seven of 40 (67.5%) of eligible trainers responded, 26 for the whole duration of the study. We feel this provided a representative sample of the horses of interest for the time period of the study. Our data provides statistics on which to base sample size calculations for future’

This information needs to be in M&M / Discussion

Line 175 A rogue extra ’.’ has crept in

line 201 degenerative

Tables 2 and 3 have not been corrected!

Incorrect spellings in Table 2 or incorrect nomenclature

Tautology

check ligament  - ALDDFT

sesamoidian - sesamoidean

ligament desmitis  - desmitis

rhabdomyolisis    - rhabdomyolysis

Incorrect spellings in Table 3 or incorrect nomenclature

Tautology

check ligament

sesamoidian

ligament desmitis    

rhabdomyolisis

third phalanx - distal phalanx

Line 423-5 Subset of the Australian racing industry – results not necessarily globally applicable

Small data set

References: Why are some journal names abbreviated and others are not? Please comply with this journal's requirements

Author Response

Please see attached word document.

I apologise for this, as most of these issues were already addressed, but as it turns out, the changes that I had made were not saved in the uploaded word documents (main document and response) and excel tables. Ana had emailed me to say that it appeared that I had uploaded the wrong document, but I had attributed this to the problems I had experienced with the MDPI portal and emailed my word document without checking it further. I sincerely apologise for this mistake. 
